# Diagnosis and Management of Central Nervous System Cryptococcal Infections in HIV-Infected Adults

**DOI:** 10.3390/jof5030065

**Published:** 2019-07-19

**Authors:** Caleb Skipper, Mahsa Abassi, David R Boulware

**Affiliations:** Department of Medicine, University of Minnesota, Minneapolis, MN 55455, USA

**Keywords:** *Cryptococcus*, cryptococcal meningitis, AIDS, HIV, antifungal therapy

## Abstract

Cryptococcal meningitis persists as a significant source of morbidity and mortality in persons with HIV/AIDS, particularly in sub-Saharan Africa. Despite increasing access to antiretrovirals, persons presenting with advanced HIV disease remains common, and *Cryptococcus* remains the most frequent etiology of adult meningitis. We performed a literature review and herein present the most up-to-date information on the diagnosis and management of cryptococcosis. Recent advances have dramatically improved the accessibility of timely and affordable diagnostics. The optimal initial antifungal management has been newly updated after the completion of a landmark clinical trial. Beyond antifungals, the control of intracranial pressure and mitigation of toxicities remain hallmarks of effective treatment. Cryptococcal meningitis continues to present challenging complications and continued research is needed.

## 1. Introduction

*Cryptococcus neoformans* is the primary etiological agent of cryptococcal disease and is a leading cause of AIDS-related meningitis. Infection occurs through inhalation of the fungus from an environmental source, typically soil contaminated by bird droppings or decaying wood [1]. Cryptococcosis is a disease that spans the globe though the greatest burden is found in sub-Saharan Africa [2]. Cryptococcal meningitis most commonly occurs in HIV-infected persons with CD4^+^ T cell counts <100 cells/μL [3]; however, despite increasing efforts to expand antiretroviral therapy (ART) access throughout sub-Saharan Africa in recent years, the incidence of cryptococcal meningitis has only minimally declined [4,5]. Acute mortality rates approach ~20% in high-income countries, ~40% in research studies in Africa, and are often higher in areas without reliable access to amphotericin [5]. Given the widespread epidemiology and substantial morbidity and mortality associated with cryptococcosis, the disease should be viewed as a major health concern requiring expanded research and focused education in management, particularly in those with advanced HIV disease. This review will provide the foundational knowledge in the clinical aspects of cryptococcal meningitis and update the readers on recent key developments in the field.

## 2. Diagnosis of Cryptococcal Meningitis

Important advancements have been made in the diagnosis of cryptococcal meningitis in the last decade, though traditional microbiology laboratory modalities continue to have a role. The isolation of *Cryptococcus* yeast on culture has been the historical gold standard for diagnosis. *Cryptococcus spp.* grow on numerous types of media although fungal isolation media such as Sabouraud’s dextrose agar is commonly used. The yeast may take 3–7 days to grow even under proper culturing conditions, making definitive diagnosis slow. Low fungal burdens can cause false negative cultures, and rapid-growing environmental fungi can out-compete *Cryptococcus* on the culture plate if contaminated. Despite these drawbacks, culture remains valuable in confirming the clearance of yeast from the cerebrospinal fluid (CSF) after induction therapy, or in distinguishing the relapse of cryptococcal meningitis from paradoxical immune reconstitution inflammatory syndrome (IRIS) [6]. In the research setting, a quantitative culture technique allows the serial measurement of fungal colony forming units (CFUs) from the CSF [7]. In one study, a fungal burden of >100,000 CFU/mL at time of cryptococcal meningitis diagnosis was associated with increased mortality [8]. Serial quantitative CSF cultures can provide clinicians highly valuable information. These serial cultures generally have log_10_-linear clearance; and a calculated value, known as early fungicidal activity (EFA), of the log_10_ CFU per mL CSF per day yeast clearance correlates with drug regimen potency, making EFA particularly useful in efficacy endpoints of phase II trials. Mortality is consistently higher at EFAs worse than 0.20 log_10_ CFU/mL/day [9]. 

India ink staining can be used for direct visualization of the yeast in CSF. Because *Cryptococcus* has a thick polysaccharide capsule, the translucent capsule highlights the yeast against the dark black ink background when viewed under a light microscope. India ink was developed to provide a timely diagnostic test for cryptococcal meningitis as compared to standard culture. However, India ink staining has sub-optimal sensitivity for the detection of *Cryptococcus* from CSF in HIV-infected persons, with studies reporting values from 50–86% [10,11]. Further, the sensitivity of India ink can be particularly poor (~40%) for persons with a fungal burden <1000 CFU/mL of CSF [11]. India ink remains a common rapid diagnostic in resource-limited settings. In HIV-infected adults in sub-Saharan Africa, the most common diagnosis in India ink-negative CSF remains cryptococcal meningitis [12]. In HIV-seronegative persons with cryptococcosis, the fungal burden is lower, and India ink is often negative. 

The detection of *Cryptococcus* via antigen testing has become an essential tool in the armamentarium of cryptococcal meningitis diagnostics. Cryptococcal antigen (CrAg) can be detected in the serum, plasma, and whole blood of those with disseminated disease; and in the CSF of those with cryptococcal meningitis. CrAg testing has become the new gold standard for diagnosing cryptococcal meningitis and is the recommended first-line approach of the WHO following lumbar puncture [13]. CrAg testing can be performed using latex agglutination, enzyme immunoassay, and lateral flow assay (LFA). The CrAg LFA (Immy Inc., Norman, Oklahoma, USA) was first FDA-approved in July 2011 and has greatly simplified CrAg testing. Whereas latex agglutination and enzyme immunoassay techniques both require laboratory personnel and user expertise to perform, the CrAg LFA can be done at the bedside with minimal training. The CrAg LFA detects cryptococcal capsule antigen using gold-conjugated monoclonal antibodies that bind the antigen and create a “sandwich” with fixed antibodies on the assay strip, resulting in color production and a line to form on the test. The CrAg LFA is stable at room temperature and does not require cold chain storage, unlike the CrAg latex agglutination tests.

The CrAg LFA has been validated as a highly sensitive and specific test for the diagnosis of cryptococcal disease. A multisite study in Uganda and South Africa demonstrated a CrAg LFA sensitivity of 99.3% and specificity of 99.1% in CSF for the diagnosis of cryptococcal meningitis [11]. In retrospect, the specificity of only 99% was likely due to the improved sensitivity of the CrAg LFA compared to reference methods. CrAg LFA has excellent performance in serum or plasma [11]. Urine and saliva have been assessed, but both specimen types were problematic, yielding false positives [14,15]. CrAg LFA is also accurate in detecting cryptococcal antigen from whole blood finger prick, although a South African study demonstrated improved sensitivity if a transfer pipette was used in asymptomatic persons with low titer levels [16]. Fingerstick testing by LFA has 100% concordance with serum or plasma LFA results, and 100% negative predictive value for excluding cryptococcal meningitis [17]. Therefore, the point-of-care LFA provides a rapid test for ruling out cryptococcal meningitis at the bedside. There are at least four other non-FDA-approved CrAg LFAs that have been developed and are being sold internationally. Clinical validation testing has been limited (or none at all), and substantial problems in specificity have been reported with at least one non-FDA-approved CrAg LFA [18]. Thus, while the CrAg LFA manufactured by Immy is the new gold standard diagnostic test, not all lateral flow assays are created equal. 

Semi-quantitative CrAg titers can be performed in the laboratory to measure the antigen burden. CrAg titer has been demonstrated to correlate with pretreatment quantitative cultures, and CrAg LFA titers ≥1:1024 are associated with greater mortality at two and ten weeks [19]. CrAg LFA has an analytical sensitivity 4–5× greater than latex agglutination, thus titers are not equivalent (i.e., a 1:1024 titer by LFA is approximately equivalent to 1:4048 titer by latex agglutination). Additionally, titers are not comparable across manufacturers. Unfortunately, performing CrAg titers can be labor intensive and require extra diluent and laboratory materials, thereby increasing the cost. A new point-of-care semi-quantitative LFA has been developed and is currently undergoing validation. This new test uses a three-line system to not only give a qualitative (positive or negative) result, but also indicates whether the antigen burden is low positive or high positive. An accurate point-of-care semi-quantitative test to delineate CrAg LFA titers ≥1:160 would prove particularly useful for identifying those with cryptococcal antigen in peripheral blood (i.e., antigenemia), who are at increased risk for progression to meningitis [20]. 

Non-specific markers may also be used to in the evaluation of persons with cryptococcal meningitis. High levels of 1,3-β-d-glucan were observed in the CSF of a cohort of Ugandan and South African HIV-infected patients with cryptococcal meningitis [21]. The sensitivity of 1,3-β-d-glucan for the detection of *Cryptococcus* was 89% in CSF and 79% in serum [21]. While not as sensitive or specific as CrAg, beta-d-glucan could be used in monitoring treatment response or in helping differentiate relapse from IRIS, as the beta-d-glucan levels decline rapidly after initiating antifungal therapy. The presence of CSF 1,3-β-d-glucan in cryptococcal meningitis contradicts the dogma that *Cryptococcus* does not produce 1,3-β-d-glucan [21]. The FDA required black box warning stating that *Cryptococcus* makes only very low levels of 1,3-β-d-glucan is incorrect [21]. CSF lactate may have some prognostic value for disease severity, but thus far research is limited. 

Molecular techniques are an emerging option for meningitis diagnosis. The Biofire Filmarray system (BioFire Diagnostics, Salt Lake City, UT) includes a multiplex PCR for 14 common pathogens isolated from the CSF. In an initial validation study, 39 persons living with HIV and with suspected first-episode cryptococcal meningitis had CSF tested by Biofire Filmarray Meningitis Encephalitis Panel [22]. The FilmArray system demonstrated good sensitivity and specificity to detect *Cryptococcus* in the CSF at high fungal burdens (96% sensitivity at ≥100 CFU/mL CSF); however, sensitivity was only 50% at low fungal burdens, <100 CFU/mL [22]. Thus, a negative PCR does not exclude cryptococcosis, and this is especially true in non-HIV populations who have lower fungal burdens. The cost and necessary laboratory facilities can be prohibitive to the wide-spread use of PCR testing in resource-limited settings. Mass spectrometry has also been used to successfully identify *Cryptococcus* from the CSF, but it is not routinely used in clinical practice [23].

## 3. Management of Cryptococcal Meningitis

The management of cryptococcal meningitis remains challenging due to the high acute mortality of the disease and the limited effective antifungal options available. Research in recent years has made strides in understanding how adjunctive therapies and supportive care can improve outcomes. Multiple clinical trials are either recently completed or currently underway to establish improved antifungal regimens, focusing primarily on maintaining efficacy while increasing accessibility and decreasing toxicity. We present the most up-to-date evidence here. There are three key principles: (1) antifungal therapy, (2) control of intracranial pressure, and (3) supportive care. All three are equally important to optimize survival. 

The first stage in the management of HIV-associated cryptococcal meningitis is the induction phase, where patients are hospitalized and emphasis is placed on rapid killing of the fungus and the management of intracranial pressure. The consolidation phase occurs after the acute hospitalization and includes the period antifungals are de-escalated and ART is initiated. The maintenance phase exists for the prevention of cryptococcal disease relapse while the individual remains immune compromised and at risk. Therefore, the three different phases of induction, consolidation, and maintenance comprise the core components in the management of cryptococcal meningitis.

### 3.1. Induction Therapy

Prior to 2018, the WHO-recommended first-line induction regimen was 14 days of intravenous amphotericin B (0.7–1.0 mg/kg per day) combined with flucytosine 100 mg/kg/day [24]. Day et al. established the efficacy of amphotericin B with flucytosine in a randomized clinical trial comparing 4 weeks of amphotericin B monotherapy, 2 weeks of amphotericin B with fluconazole, and 2 weeks of amphotericin B with flucytosine. The combination of amphotericin B with flucytosine was associated with ~40% lower mortality risk at 10 weeks than amphotericin monotherapy [7]. In Africa, the combination of amphotericin B with high-dose fluconazole (800–1200 mg per day) for 14 days is a commonly used alternative due to the difficulty in obtaining flucytosine [5,25,26,27].

The WHO issued updated guidelines in 2018 on the management of cryptococcal meningitis, in which they changed the preferred induction regimen to 7 days of intravenous amphotericin B with flucytosine, followed by 7 days of high dose fluconazole (1200 mg per day) (Table 1) [28]. The new recommendations came following completion of the Antifungal Combinations for Treatment of Cryptococcal Meningitis in Africa (ACTA) trial [29]. ACTA was a phase III, open-label, randomized trial conducted at multiple sites throughout Africa. A total of 721 participants were enrolled and randomized to one of: 2 weeks of amphotericin B, 1 week of amphotericin B, or a 2-week all-oral regimen of high dose fluconazole plus flucytosine. The amphotericin B arms were paired with either fluconazole or flucytosine. The five arms were compared against each other for all-cause mortality at 2 and 10 weeks. Ten-week mortality rates were lowest in the one week of amphotericin B with flucytosine arm (24%) and all-oral fluconazole plus flucytosine (35%) arm (Figure 1) [29]. Conversely, the highest mortality was seen in the one week of amphotericin B with fluconazole arm (41.3%). Overall, amphotericin B paired with flucytosine was superior to amphotericin B paired with fluconazole.

Unfortunately, flucytosine remains either unavailable or prohibitively costly through much of the world, despite its superior efficacy. Efforts are currently underway in 2019–2020 by UNITAID to make flucytosine more readily available. In Europe, flucytosine is modestly priced ($25/day), while in the United States, flucytosine costs $425/day in 2019 down from $2000/day in 2016 [30]. 

Studies of novel adjunctive therapies have been carried out in search of regimens to further reduce mortality from cryptococcal meningitis. Interestingly, the antidepressant sertraline has good in vitro activity against *Cryptococcus* [31] and achieves good concentrations in the brain [32], making it an attractive target as an adjunct to standard antifungal therapy. Unfortunately, this did not translate into any survival benefit with the addition of sertraline over amphotericin and fluconazole in a randomized trial [33]. Similarly, the selective estrogen-receptor modulator tamoxifen has in vitro activity against *Cryptococcus* and has synergy with amphotericin B [34], but this may not improve EFA.

Interferon-gamma (IFN-γ) is an adjunctive therapy of interest in the management of human cryptococcal disease. IFN-γ plays a critical role in the CD4^+^ helper T cell response, an essential component of the host’s defense against *Cryptococcus.* CD4^+^ helper T cells can be directed toward a predominant Type 1 (Th1) or Type 2 (Th2) response characterized by their cytokine profiles; IFN-γ is associated with a Th1 response. In animal models, Th1 responses lend an effective adaptive immune response to *Cryptococcus* [35,36,37], whereas Th2 responses, classically geared toward allergic and parasitic stimuli, appear detrimental in cryptococcal infection [38,39,40]. Jarvis et al. demonstrated that HIV-infected persons with cryptococcal meningitis whose CSF was predominated by IFN-γ or tumor necrosis factor-alpha had improved two-week survival [41]. A randomized trial in South Africa demonstrated a 30% increase in the rate of yeast clearance from the CSF when amphotericin B and flucytosine were combined with two doses of IFN-γ compared to antifungals alone [42], although there was no overt survival benefit. IFN-γ remains an intriguing option in the management of cryptococcal meningitis and further study is warranted. 

Current evidence does not support the use of adjunctive corticosteroids for the treatment of HIV-associated cryptococcal meningitis, in contrast to the benefit seen with steroids in bacterial and tuberculous meningitis [43,44]. A multisite trial of HIV-associated cryptococcal meningitis found increased rates of infectious, renal, and cardiovascular adverse events in persons randomized to dexamethasone versus placebo, with no mortality benefit at 6 months [45]. Corticosteroids may be useful in *Cryptococcus*-associated paradoxical IRIS [46], though evidence is limited and management decisions are often based on clinical judgement. 

For the first time in many years, novel antifungals are entering the pipeline as possible therapeutics for cryptococcal meningitis. Multiple phase I and phase II clinical trials will be investigating novel agents or new formulations in 2019–2021, including an orally bioavailable form of amphotericin B and a new small-molecule antifungal from a novel class of glycosylphosphatidylinositol (GPI)-anchored wall transfer protein 1 (Gwt1) inhibitors blocking the transport of the fungal mannoprotein from the Golgi complex to the cell wall. Additional ongoing trials are investigating the use of liposomal amphotericin B at 10 mg/kg as a single dose both for the treatment of meningitis (AMBITION-CM Trial) and preemptive treatment of CrAg antigenemia (ACACIA Trial). Results of the AMBITION-CM Trial are expected by early to mid-2021. 

### 3.2. Consolidation Therapy

The consolidation phase of therapy consists of fluconazole 400–800 mg/day through at least 10 weeks [13,47]. Typically, consolidation therapy is started two weeks after induction therapy, although the transition to the consolidation phase should be based on the individual patient’s response to induction therapy. Even with approved induction regimens, the CSF does not always sterilize by the end of the first two weeks. A Ugandan study found 56% of patients treated with amphotericin combination therapy with fluconazole had positive cultures at the end of two weeks [5]. Fluconazole at 400 mg/day is fungistatic [48], and persons with positive 2-week cultures have an increased risk of 10-week mortality when receiving 400 mg/day [49]. Therefore, a lumbar puncture and culture should be performed at two weeks to prove eradication of viable yeast from the CSF, before reducing the fluconazole dose. When continuing fluconazole at 800 mg/day until ART is initiated and the CSF is known to be sterile, there was no increased mortality risk with a positive 2-week cultures [50]. Thus, fluconazole 800 mg/day in split doses during consolidation phase has become the recommendation [28]. Longer durations of fluconazole at higher doses during the consolidation phase should be considered if using suboptimal induction therapy, particularly fluconazole monotherapy or when CSF sterility has not been achieved [47]. 

Itraconazole is an alternative for consolidation therapy, but it is not recommended given slightly worse efficacy compared with fluconazole and increased concern of drug-drug interactions, particularly with antiretrovirals [49,51]. Some microbiology laboratories may report *Candida* susceptibility (≤4 mcg/mL) breakpoints for *Cryptococcus*. This may result in unwarranted switches away from fluconazole. 

### 3.3. Maintenance Therapy

Immunocompromised persons convalescing from recent cryptococcal meningitis remain at risk for disease until a critical level of immune reconstitution is achieved. Fluconazole 200 mg daily has proven effective in preventing the recurrence of cryptococcal disease as maintenance therapy in ART-naïve persons [52]. Both once-weekly amphotericin B and daily itraconazole had higher rates of culture-positive relapse compared to daily fluconazole when used as long-term maintenance therapy to prevent the recurrence of cryptococcal meningitis [53,54]. In the setting of elevated fluconazole MICs above 16 μg/mL, one should double the secondary prophylaxis dose for every two-fold MIC increased thereafter (e.g., the fluconazole minimum inhibitory concentration (MIC) of 32 μg/mL => 400 mg/day fluconazole secondary prophylaxis). The WHO criteria for discontinuing fluconazole maintenance therapy require that, “the person is stable on and adherent to ART and antifungal maintenance treatment for at least one year and has a CD4 cell count ≥100 cells/μL and a fully suppressed viral load [28].” When HIV viral load testing is unavailable, discontinuation if CD4 counts are >200 cells/μL after one year of maintenance therapy is instructed.

### 3.4. Management of Intracranial Pressure 

The second key principle is control of elevated intracranial pressure as 50–75% will present with elevated intracranial pressure (ICP) of >20 cm H_2_O. *Cryptococcus* is thought to cause elevated ICP as the yeast’s “sticky” polysaccharide capsule coats the arachnoid villi and physically obstructs CSF resorption [55]. Elevated ICP is clinically characterized by headache, vomiting, confusion, papilledema, and loss of visual acuity [56,57]. Of note, cranial nerve palsies can occur, with cranial nerve VI involvement commonly described [58]. However, it should be noted that raised ICP may be present in the absence of overt symptoms [59]. 

Bicanic et al. demonstrated that baseline opening pressure correlated with quantitative cryptococcal fungal burden [60], while high baseline fungal burden has known association with increased mortality [61]. Accordingly, studies have demonstrated clinical benefits, including survival, with aggressive ICP management in HIV-associated cryptococcal meningitis. A study involving Ugandan and South African participants found a 69% relative reduction in 10-day mortality in those who received at least one therapeutic lumbar puncture in the first week versus those who did not, regardless of baseline opening pressure (Figure 2) [62]. The WHO recommends that therapeutic lumbar punctures be performed as needed to reduce intracranial pressure to 20 cm H_2_O or less, and to use symptoms to guide the frequency of CSF drainage [28]. The use of pharmacotherapeutics, such mannitol, acetazolamide, or corticosteroids, are not recommended for managing elevated ICP in HIV-associated cryptococcal meningitis [63,64,65]. 

In many scenarios, CSF opening pressure may not be initially measured. In this event, the strong recommendation is to repeat a lumbar puncture within 24 h to measure the opening pressure. In the absence of manometer, removal of ~20mL of CSF is recommended by these authors. 

### 3.5. Management of Amphotericin B-Related Toxicities 

The third key principle is supportive care. Amphotericin B deoxycholate has well known toxicities, and managing these toxicities is critically important with a survival benefit [66]. In high-income countries, liposomal or lipid formulations of amphotericin are recommended to decrease toxicity [67]. 

The toxicities of amphotericin include infusion reactions, renal dysfunction, anemia, and electrolyte abnormalities. Amphotericin B causes the release of pro-inflammatory cytokines such as IL-1β, IL-6, and MIP-1β as it is recognized as a microbial by-product (produced from *Streptomyces spp.*) by host toll-like receptors [68]. Infusion reactions include malaise, fevers, and rigors; they are typically self-limited. Acetaminophen can be given for symptomatic management with hydrocortisone reserved for severe infusion reactions [69]. 

Amphotericin B causes rapid vasoconstriction of the afferent renal arterioles, resulting in a decrease in glomerular filtration rate [70]. Acute kidney injury occurs with cumulative doses of amphotericin but is reversible with discontinuation. The administration of intravenous fluids pre- and post- infusion is used to reduce the risk of renal injury. The WHO recommends increasing fluid administration to 1 liter every eight hours or alternating daily dosing of amphotericin B if there is a greater than two-fold rise in serum creatinine from baseline [65]. Importantly, the fungicidal benefit of longer durations of amphotericin B may be outweighed by the risk of substantial toxicity; the traditional paradigm of 14-day therapies may be unfavorable compared to shorter courses. 

Potassium and magnesium loss invariably occurs with amphotericin B administration and are related to renal tubular damage. Close electrolyte monitoring is critical, and supplementation is often necessary to prevent life-threatening hypokalemia. If electrolyte monitoring is not readily available, a standardized protocol for electrolyte supplementation should be instituted. The benefit of standardized electrolyte supplementation was demonstrated by Bahr et al. where the cumulative incidence of severe hypokalemia (<2.5 mEq/L) decreased from 38 to 8.5% with a universal electrolyte supplementation package begun on day 1 [66]. This package consisted of 40 mEq/day KCl during week one and 60 mEq/day during week two of amphotericin coupled with 8 mEq/day of Mg^++^. Current guidelines recommend monitoring potassium levels 2–3 times weekly, particularly in the second week of therapy [65]. 

Anemia occurs primarily from amphotericin B induced erythropoietin suppression but can also result from hemolysis [71]. Hemoglobin levels may drop by a mean of 1.5 g/dL following 7 days of amphotericin B therapy in South Africa and up to 4 g after 14 days in Uganda, and up to one-third may develop anemia worse than 8.5 g/dL [72,73]. Hemoglobin levels should be monitored weekly and as clinically indicated. Anemia decreases oxygen carrying capacity to the brain, resulting in a strong risk factor for mortality [73,74]. Red blood cell transfusions may be a critical, life-saving intervention. “Weakness” at the conclusion of two weeks of amphotericin can be misattributed to anemia in the presence of unknown hypokalemia or hypomagnesemia. 

### 3.6. ART Initiation

The 2017 WHO Guidelines for Managing Advanced HIV Disease and Rapid Initiation of Antiretroviral Therapy emphasize early and rapid ART initiation in persons living with HIV, including those with advanced disease [75]. Rapid ART initiation has been linked to several benefits, including improved retention in care, viral suppression, and mortality, at 12 months [76]. However, these benefits must be weighed against possible competing harms of ART initiation in those with opportunistic infections at risk for IRIS. 

In a landmark trial, ART naïve HIV-infected participants with first-episode cryptococcal meningitis randomized to initiate ART at 1–2 weeks from diagnosis had 15% higher 26-week mortality than those initiating ART at 4–6 weeks [8]. ART initiation is recommended after sustained clinical response and after four weeks of effective amphotericin-based treatment. Although robust data are lacking, the same practice is applied to ART switches in ART-experienced persons presenting with cryptococcal meningitis. 

## 4. Immune Reconstitution Inflammatory Syndrome versus Cryptococcal Meningitis Relapse

Immune reconstitution inflammatory syndrome is an exaggerated and dysregulated immune response to an opportunistic pathogen upon restoration of a deplete component of the immune system. IRIS can range in severity from mild illness to life-threatening disease, depending on the pathogen and organ system involved. In cryptococcal meningitis, the excess inflammation in the closed space of the skull can be fatal. The reported incidence of paradoxical cryptococcal IRIS is highly variable in incidence, ranging between 8–49%, presenting as soon as 4 days and up to 6 years after ART initiation, and carrying a mortality rate of 0–36% [77,78,79].

Recurrent meningitis symptoms after the treatment of cryptococcal meningitis and post ART initiation raises suspicion for the development of paradoxical IRIS, versus relapse of the cryptococcal meningitis. A damage-response framework model has been proposed whereby disease can occur secondary to either uncontrolled pathogen dissemination or the host’s own immune system (Figure 3) [80]. Distinguishing paradoxical IRIS from cryptococcal meningitis relapse is often difficult and assessing fluconazole adherence is an important consideration. A sterile culture and high CSF white cell count support the diagnosis of paradoxical IRIS [77,81]. Conversely, positive cultures, particularly in the setting of poor fluconazole adherence, suggests cryptococcal meningitis relapse. In the setting of persistently positive cultures and a history of good fluconazole adherence, treatment failure is likely and the cryptococcal isolate should undergo susceptibility testing to fluconazole. Fluconazole resistance may be contributing when the minimum inhibitory concentration (MIC) is ≥32 μg/mL. Itraconazole or voriconazole has been used as salvage therapy in cases of fluconazole resistance; however, both of these have substantial drug-drug interactions with ART [82]. In the setting of fluconazole resistance, and limited alternatives, re-induction with amphotericin B to achieve sterility is imperative, followed by higher dose fluconazole (i.e., 1200 mg/day) with consideration of weekly amphotericin B. 

The management of IRIS is largely based on expert opinion and individual case factors. Like cryptococcal meningitis, elevated intracranial pressure is detrimental in IRIS [83], and management with therapeutic lumbar punctures is critical. The use of corticosteroids for the management of paradoxical IRIS is controversial. The 2010 IDSA guidelines recommend 0.5–1 mg/kg of prednisone or dexamethasone to be tapered over a 2–6-week period in severe cases [47]. The WHO guidelines only state that corticosteroids may be considered. Neurologic improvement with the use of the TNF-α inhibitors, thalidomide and adalimumab, has been reported [84,85].

## 5. Special Situations

### 5.1. Pregnancy

Pregnancy presents a challenging scenario for the treatment of cryptococcal meningitis due to fluconazole and flucytosine both having concerns of teratogenicity, particularly in the first trimester [86,87]. Pregnant women are consistently excluded from research studies involving *Cryptococcus*, and little data exist on clinical outcomes in pregnancy and cryptococcal disease. Pregnant women presenting with cryptococcal meningitis should be treated with amphotericin B (FDA Pregnancy Category B) for induction therapy and may need continued amphotericin B weekly throughout the consolidation period when in the first trimester. Clinical judgement and patient counseling are required for the consideration of fluconazole after the first trimester, where benefits must be weighed against risks. Outside of the first trimester, the absolute risks of fluconazole adverse events are lower but not zero. The use of fluconazole as maintenance therapy in women of child-bearing age is also an area of concern where more research is needed. 

### 5.2. Cryptococcus neoformans versus C. gattii

The vast majority of HIV-associated cryptococcosis is secondary to *Cryptococcus neoformans*, but *Cryptococcus gattii* is another pathogenic species. Interestingly, *C.gattii* has greater propensity to cause disease in immunocompetent persons, becoming apparent during an initial outbreak in the Pacific Northwest of North America. Currently, the clinical management of cryptococcal meningitis is the same for either species. As distinguishing the two species requires special media or molecular testing, specification is rarely performed in low- or middle-income countries. 

### 5.3. Unmasking Disease

Unmasking cryptococcal disease is a clinical phenomenon where subclinical disease becomes symptomatically manifest after the initiation of ART. “Test and treat” strategies are becoming commonplace throughout sub-Saharan Africa, and ART-experienced first-episode cryptococcal meningitis is being seen more frequently. In a Ugandan study of first-episode cryptococcal meningitis, 46% of participants were receiving ART at presentation [88]. Of those, two-week mortality was 47% in those receiving ART for ≤14 days, compared with 19% in those receiving ART for 15–182 days and 26% in those receiving ART for >6 months. It is currently unknown if ART should be continued in these patients. Unmasking disease is an area in need of more research. 

### 5.4. Cerebral Cryptococcoma

Cerebral cryptococcoma, a rare complication of cryptococcal disease, is a localization of *Cryptococcus* in the brain parenchyma. Cryptococcomas are poorly understood, with most literature being limited to case reports. They are likely more common in *Cryptococcus gattii* infections in immunocompetent persons [89], though certainly HIV-associated cases are described [90]. In general, cryptococcomas are exceedingly rare in HIV-infected ART-naïve persons; however, unmasking cryptococcoma(s) can occur after ART initiation in untreated or ineffectively treated infections [91,92]. Cryptococcoma(s) can appear as ring-enhancing lesions on head CT and be confused with cerebral toxoplasmosis. No prospective studies are available to inform optimal management. IDSA guidelines recommend induction therapy with amphotericin B and flucytosine for at least 6 weeks, followed by fluconazole consolidation and maintenance for up to 18 months [47], but acknowledge these recommendations are based on limited evidence. We recommend an alternative approach of re-induction with amphotericin, verification of fluconazole susceptibility, and then use of higher dose fluconazole at 1200 mg/day. If the cryptococcoma is causing cerebral edema and mass effect, then corticosteroids and surgery can be considered.

### 5.5. Asymptomatic Antigenemia

The topic of asymptomatic cryptococcal antigenemia is beyond the scope of this review of central nervous system cryptococcal infection, though much literature is available on the topic, including a recently published prospective cohort study [93]. Briefly, severely immunocompromised persons without any symptoms of meningitis may have detectable cryptococcal antigen in their blood. Asymptomatic antigenemia should be seen as a continuum of disease, where *Cryptococcus* has been able to escape immune control in the lung and can disseminate but is not yet causing symptomatic meningitis. However, persons with cryptococcal antigenemia are at high risk of progression to meningitis. The use of pre-emptive fluconazole improves meningitis-free survival [91,94,95,96]. A randomized controlled trial using high-dose liposomal amphotericin with fluconazole as pre-emptive therapy in persons with cryptococcal antigenemia is currently underway.

## 6. Summary

Cryptococcal meningitis remains a significant source of morbidity and mortality in those with advanced HIV disease, particularly in lower-resource settings. The proper management of opportunistic infections starts with diagnosis, and the advent of the CrAg LFA as a means to accurately and rapidly diagnose cryptococcal disease is immensely valuable. Culture is still important for determining relapse from IRIS and allows for the calculation of EFA as an indicator of efficacy in research trials. Molecular methods hold promise but remain limited by practical factors. 

The combination of amphotericin B and flucytosine is the current standard for induction therapy. Consolidation and maintenance therapies rely on long-term adherence to fluconazole, and early discontinuation can lead to cryptococcal meningitis relapse. Adjunctive therapies to induction antifungal regimens have been explored but are so far without convincing benefit. Supportive care with therapeutic lumbar punctures, fluid administration, and electrolyte supplementation are critical to optimizing good outcomes. ART should be initiated 4–6 week after the diagnosis of cryptococcal meningitis. Clinicians should be cognizant for the signs and laboratory findings of paradoxical IRIS.

## Figures and Tables

**Figure 1 jof-05-00065-f001:**
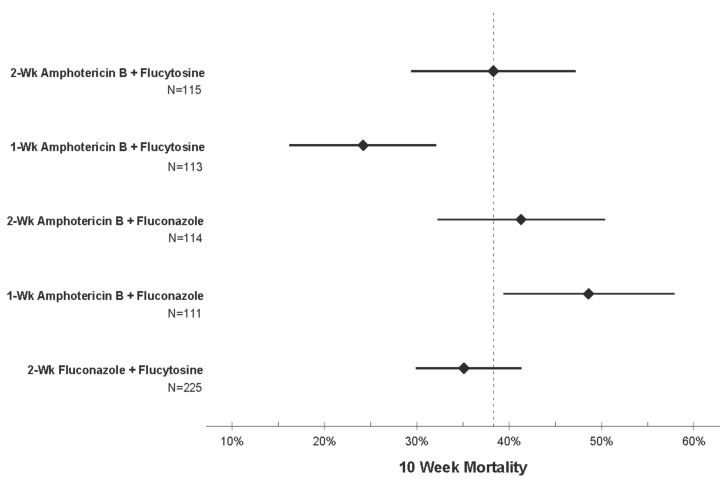
Ten-week mortality by antifungal regimen for cryptococcal meningitis from the Antifungal Combinations for Treatment of Cryptococcal Meningitis in Africa (ACTA) trial [29]. The ACTA trial compared 10-week mortality among five antifungal regimens. The dotted line represents the mean survival of the comparison group (two weeks of amphotericin B plus flucytosine). Mean survival times are noted by the diamonds with the 95% confidence intervals represented by the error bars. One week of amphotericin B with flucytosine demonstrated the lowest mortality of the group, likely due to less amphotericin-related toxicity combined with the improved efficacy of flucytosine over fluconazole.

**Figure 2 jof-05-00065-f002:**
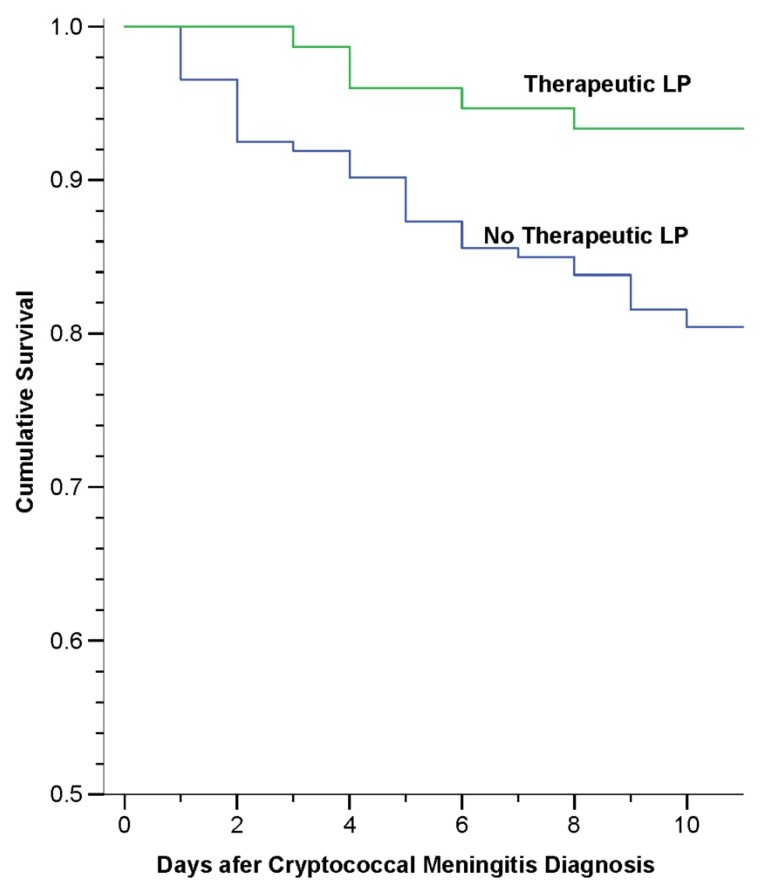
Ten-Day Survival by Receipt of at Least One Therapeutic Lumbar Puncture (LP) [62]. Figure 2 is adapted from Rolfes et al. [62] to display the mortality with and without a therapeutic LP conducted during the first week, limited to those who survived >1 day. The median time to therapeutic LP was 3 days (IQR, 2–4 days). Thirty-one deaths (18%) occurred among 173 individuals without a therapeutic LP and 5 deaths (7%) among 75 with at least 1 therapeutic LP.

**Figure 3 jof-05-00065-f003:**
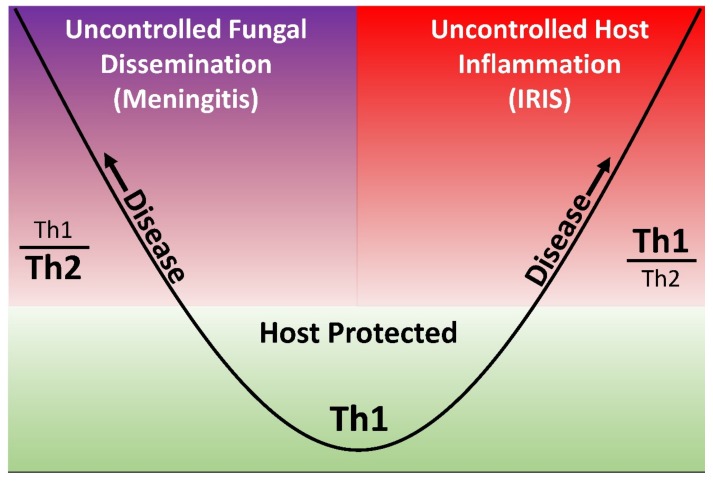
Damage-response parabolic framework in cryptococcal meningitis (adapted from Pirofski and Casadevall [80]). Figure 3 demonstrates a parabolic model of damage-response, whereby an immune response characterized by dysregulated Th2 activity is hypothesized to promote fungal dissemination in the absence of an effective immune response. Conversely, a response characterized by exuberant Th1 activity may result in clinical disease with excessive, pathogenic host inflammation, such as occurs with immune reconstitution inflammatory (IRIS).

**Table 1 jof-05-00065-t001:** The 2018 WHO Antifungal Treatment Recommendations for Cryptococcal Meningitis [28].

Medication and Dose	Week 1	Week 2	Week 3–10	Week > 10
Amphotericin B (1.0 mg/kg/day) + flucytosine 100 mg/kg/day	X ^a^			
Fluconazole 1200 mg daily		X		
Fluconazole 800 mg daily			X	
Fluconazole 200 mg daily				Through 12 months
Treatment Phase	**Induction Therapy**		**Consolidation**	**Maintenance**

WHO-recommended first-line antifungal therapy for induction, consolidation, and maintenance phases for the treatment of cryptococcal meningitis. ^a^ Therapeutic lumbar punctures and electrolyte supplementation are most critical during this period.

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
