# Peer review of "Diagnosis and Management of Central Nervous System Cryptococcal Infections in HIV-Infected Adults"

_jof, 2019, doi:10.3390/jof5030065_

Reviewer 1 Report

Skipper and colleagues reviewed current methods for diagnosis and management of CNS cryptococcal infections in HIV positive individuals. This manuscript is fantastic and will be especially useful for basic science researchers and individuals non-related with the field to understand the clinical management of cryptococcal meningitis. There are just a few suggestions for the authors:

Please consider to modify the title to "Diagnosis and management of CNS cryptococcal infections in HIV-infected adults" to make it more informative.

IRIS is defined multiple times and it should be defined only once.

In the asymptomatic antigenemia section, can the authors provide a brief description of the impact of having cryptococcal antigen in the blood?

Author Response

Please consider to modify the title to "Diagnosis and management of CNS cryptococcal infections in HIV-infected adults" to make it more informative.

-- Done!

IRIS is defined multiple times and it should be defined only once.

-- Thank you for the good observation, corrected.

In the asymptomatic antigenemia section, can the authors provide a brief description of the impact of having cryptococcal antigen in the blood?

-- Added some wording increase understanding of significance of antigenemia

Reviewer 2 Report

This is a timely discussion of therapy for Cryptococosis. Manuscript is well written and cites appropriate references. The only aspect that the authors have not addressed is ongoing clinical trials (AMBITION or others) and what these trials are trying to achieve: a reference to these  would point to reader to ongoing efforts to improve management of the disease and really show the state of the art.

Minor comments:

What is the gold standard to detect Cryptococcus? Is it mass spectrometry? A small phrase with a conclusion should be included. For example should CrAg LFA become the gold standard? Future directions?

Could the authors include a brief mention on how to distinguish between species ( C. neoformans and C gattii) and the clinical usefulness( or irrelevance) of distinguishing between these species?

Line 132: improved regimens or cheaper more accessible and equally as effective regimens?

 Fig1 is very nice.

Minor details: I would suggest including the adjunctive therapy and supportive in this graph. What do authors mean by secondary prophylaxis, shouldn’t it be maintenance phase?

Figure 2. Could the authors discuss why  the ACTA trial obtained better results with 1 wk AmpB_+FluC than 2wk?

Line 209: please list the new drugs in the pipeline, at least the ones past phase II of clinical trials.

Line 230-232: is this published or do the authors know this via the grapevine?

Line 237: Please discuss why is maintenance with fluconazole more effective than AmpB?

Author Response

This is a timely discussion of therapy for Cryptococosis. Manuscript is well written and cites appropriate references. The only aspect that the authors have not addressed is ongoing clinical trials (AMBITION or others) and what these trials are trying to achieve: a reference to these would point to reader to ongoing efforts to improve management of the disease and really show the state of the art.

Response: In the absence of data, it is difficult to discuss AMBITION trial which will be completed in Dec 2020 or early 2021 with results expected to be reported in mid 2021. We can mention this.

Minor comments:

Comment: What is the gold standard to detect Cryptococcus? Is it mass spectrometry? A small phrase with a conclusion should be included. For example should CrAg LFA become the gold standard? Future directions?

Response: Cryptococcal Antigen LFA (Immy) is the most sensitive assay and is the new gold standard. We will state this.  Culture still has a notable role (as noted on line 39) in some situations as CrAg can't distinguish between prior infection, relapse and IRIS.

Comment: Could the authors include a brief mention on how to distinguish between species ( C. neoformans and C gattii) and the clinical usefulness( or irrelevance) of distinguishing between these species?

Response: We have added a paragraph in the special situation section.  As there is no difference in treatment recommendations, our excitement is less.

Comment: Line 132: improved regimens or cheaper more accessible and equally as effective regimens?

Response: Adjusted wording of sentence 

Comment: Minor details: I would suggest including the adjunctive therapy and supportive in this graph. What do authors mean by secondary prophylaxis, shouldn’t it be maintenance phase?

Response: Changed secondary prophylaxis to maintenance, for more consistent language usage. Additionally added a superscript emphasizing importance of therapeutic LPs and supportive care.  

Comment: Figure 2. Could the authors discuss why  the ACTA trial obtained better results with 1 wk AmpB_+FluC than 2wk?

Response: Added to caption

Comment: Line 209: please list the new drugs in the pipeline, at least the ones past phase II of clinical trials.

Response: Included two examples of oral ampho and amplyx APX001 molecule entering in Phase II trials in 2020.

Line 230-232: is this published or do the authors know this via the grapevine?

Response: We will modify this as: “Some microbiology laboratories may report Candida susceptibility (<4 mcg/mL) as unofficial breakpoints for Cryptococcus (while stating these are not official breakpoints).” --  This happens locally at multiple independent institutions in MN (which I am slightly appalled by).

Line 237: Please discuss why is maintenance with fluconazole more effective than AmpB?

Response: Sentence updated